# Distinct immune responses associated with vaccination status and protection outcomes after malaria challenge

Damian A. Oyong[1]*, Fergal J. Duffy[1], Maxwell L. Neal[1], Ying Du[1], Jason Carnes[1], Katharine V. Schwedhelm[2], Nina Hertoghs[1], Seong-Hwan Jun[2], Helen Miller[2], John D. Aitchison[1], Stephen C. De Rosa[2], Evan W. Newell[2], M Juliana McElrath[2], Suzanne M. McDermott[1], Kenneth D. Stuart ⬚[1]*

1 Center for Global Infectious Disease Research (CGIDR), Seattle Children's Research Institute, Seattle, Washington, United States of America, 2 Vaccine and Infectious Disease Division, Fred Hutchinson Cancer Center, Seattle, Washington, United States of America

* damian.oyong@seattlechildrens.org (DAO); ken.stuart@seattlechildrens.org (KDS)

## Abstract

Understanding immune mechanisms that mediate malaria protection is critical for improving vaccine development. Vaccination with radiation-attenuated *Plasmodium falciparum* sporozoites (PfRAS) induces high level of sterilizing immunity against malaria and serves as a valuable tool for the study of protective mechanisms. To identify vaccine-induced and protection-associated responses during malarial infection, we performed transcriptome profiling of whole blood and in-depth cellular profiling of PBMCs from volunteers who received either PfRAS or noninfectious mosquito bites, followed by controlled human malaria infection (CHMI) challenge. In-depth single-cell profiling of cell subsets that respond to CHMI in mock-vaccinated individuals showed a predominantly inflammatory transcriptome response. Whole blood transcriptome analysis revealed that gene sets associated with type I and II interferon and NK cell responses were increased in prior to CHMI while T and B cell signatures were decreased as early as one day following CHMI in protected vaccinees. In contrast, non-protected vaccinees and mock-vaccinated individuals exhibited shared transcriptome changes after CHMI characterized by decreased innate cell signatures and inflammatory responses. Additionally, immunophenotyping data showed different induction profiles of vδ2+ γδ T cells, CD56+ CD8+ T effector memory (Tem) cells, and non-classical monocytes between protected vaccinees and individuals developing blood-stage parasitemia, following treatment and resolution of infection. Our data provide key insights in understanding immune mechanistic pathways of PfRAS-induced protection and infective CHMI. We demonstrate that vaccine-induced immune response is heterogenous between protected and non-protected vaccinees and that inducted-malaria protection by PfRAS is associated with early and rapid changes in interferon, NK cell and adaptive immune responses.

**Trial Registration**: ClinicalTrials.gov NCT01994525.

**Data Availability Statement:** Whole blood transcriptomic data is available on NCBI GEO under accessions GSE116619 (B0) and GSE192757 (C0, C1, C7). Flow cytometry data is available under

Supporting Information. CITE-seq data is available on Figshare https://doi.org/10.6084/m9.figshare.22697011.v1.

**Funding:** This work was supported by National Institutes of Health (https://www.nih.gov/grants-funding) grant U19AI128914 (to KDS and MJM), Bill & Melinda Gates Foundation (https://www.ghvap.org/Pages/default.aspx) grant GHVAP NG-ID18-Stuart (to KDS) and and National Institute of General Medical Sciences (https://www.nigms.nih.gov/) grant P41GM109824 to (JDA). The funders had no role in study design, data collection, analysis and in decision to publish, or preparation of the manuscript.

**Competing interests:** The authors have declared that no competing interests exist.

## Author summary

Malaria poses a significant global health threat, causing over half a million deaths annually. Effective vaccines are critically needed to prevent malaria disease. Our incomplete understanding of immune mechanisms that mediate malaria protection is hampering the development of effective vaccines. Irradiated sporozoite vaccines can induce highly sterilizing protection against malaria and are a valuable tool for the analysis of immune protection. Here, we aimed to characterize correlates of immune protection in individuals vaccinated with a suboptimal dose of irradiated sporozoite and subsequently challenged with live malaria parasite. Blood samples were taken before and after malaria challenge, and gene expression and cell type profiles were measured. We observed that the trajectories of immune response after malaria challenge is highly distinct between protected and non-protected vaccinees. We observed early perturbations in interferon, NK cell, and adaptive immune responses in protected vaccinees whereas inflammatory and innate cell response were unique to non-protected vaccinees. We also observed that the immune profile after malaria challenge was distinctly similar between non-protected vaccinees and mock-vaccinated individuals. Our study sheds light on the dynamics of vaccine-induced immune responses that are associated with protection from malaria after CHMI.

## Introduction

Malaria remains a significant threat to global health, causing an estimated 241 million cases and 627,000 deaths in 2020 [1]. Out of the five parasite species causing malaria, infection by *Plasmodium falciparum* accounts for the majority of global malaria cases and up to 95% of all cases in Africa [1]. The recent pandemic caused by severe acute respiratory syndrome coronavirus 2 (SARS-CoV2) infection has also exacerbated malaria health burdens by disrupting treatment services, increasing the number of malaria-related mortality [1]. The development of an effective anti-malaria vaccine is critically needed to prevent malaria and reduce its health burdens that disproportionally affect young children and pregnant women. To date, there is only one malaria vaccine recommended by the WHO for widespread use among sub-Saharan African children, which is the RTS,S/AS01 subunit vaccine. Nonetheless, vaccine efficacy for RTS,S/AS01 is very modest, at about 26–36% when assessed in children during a phase 3 field trial [2]. Although another similarly designed subunit vaccine showed better efficacy in a phase 2b trial, a wider age range, larger number of participants, and assessments across different regions and transmission settings are required to fully assess its effectiveness [3,4]. A promising alternative to subunit vaccines are attenuated whole *P. falciparum* sporozoite vaccines which include radiation-attenuated sporozoite (RAS); wild-type sporozoite administration under drug cover, also known as CPS (chemical prophylaxis with sporozoite) or ITV (infection treatment vaccination) [5]; and genetically attenuated sporozoite (GAP) approaches [6,7]. RAS arrests early in liver-stage development and, historically, have been highly effective in inducing sterile protective immunity in animal models and humans, with a >90% efficacy rate [5,8–11]. However, the efficacy of RAS vaccines in malaria endemic regions may be less strong, as reduced efficacy has been observed in pre-exposed individuals [12–14]. The human immune system is critical in determining the outcome of sterile immunity against malaria, and our incomplete understanding of the dynamics and mechanisms of natural or vaccine-induced immune responses to malaria is hampering the development of effective vaccines.

Identifying the mechanistic correlates of malaria protection have remained elusive, in part due to the complexity of the parasite's multi-stage life cycle and host-parasite interactions [15]. Studies of RAS vaccines have associated sterilizing protection with multiple adaptive and

innate immune cell types, including CD4+ T cells [16], CD8+ T cells [9,17–19], γδ T cells [20,21], NK cells [22], and dendritic cells (DCs) [23]. For example, abrogation of CD4+ T cells [16] and γδ T cells [21] in RAS-immunised mice resulted in a loss of malaria protection. Also, a higher frequency of CD8+ T cells secreting IFN-γ has been previously observed in individuals protected against controlled human malaria infection (CHMI) challenge compared to those who were non-protected [18]. Indeed, higher interferon-related transcription signatures have been associated with malaria protection [24]. Although the practicality of using RAS for primary vaccinations is uncertain due to logistical and large-scale manufacturing challenges [25,26], it remains a valuable tool for the analysis of immunity to malaria. For example, suboptimal dosing with RAS can produce different outcomes of protection or no protection when assessed with CHMI. This can be harnessed as an intentional strategy to investigate correlates of malaria protection.

In this study, we examined whole blood immune responses after malaria challenge in volunteers receiving *P. falciparum* RAS vaccine. Trial participants either received 5 RAS immunizations, dosed to achieve ~50% protection outcome, or uninfected mosquito bites (mock vaccination), followed by CHMI to assess malaria protection. To characterize vaccine-induced protection, we analyzed blood samples at sampling time points prior to and after CHMI and then utilized systems biology analyses of whole blood transcriptomic and cellular immune profiles. We provide evidence that malaria protected vaccinees induce distinct immune responses, including interferon and adaptive responses, and with different kinetics following CHMI compared to non-protected vaccinees and mock-vaccinated individuals.

## Results

### Study participants

Blood samples from volunteers enrolled in Cohort 1 of the Immunization by Mosquito bite with RAS (IMRAS) trial [27] were the focus of this study. Briefly, individuals in the IMRAS trial received 5 doses, at approximately monthly intervals, *P. falciparum* radiation attenuated sporozoites (PfRAS) by bites of infected mosquitoes (PfRAS vaccinees, n = 11) or non-infected mosquitoes (mock, n = 5). The number of infectious bites delivered was selected to achieve incomplete protection in the study cohort, with a target of ~50% protection outcome, to facilitate assessment of correlates of protection. An average of 1,027 infectious bites were administered to Cohort 1 over five vaccination doses. Protection from malaria was assessed with CHMI by bites from 5 mosquitoes infected with homologous *P. falciparum* strain (**Fig 1A**). Of the PfRAS vaccinated group, 5 individuals developed blood-stage parasitemia following CHMI (non-protected, n = 5) while 6 others remained negative (protected, n = 6). All mock-vaccinated individuals developed blood-stage infection after CHMI. Blood parasitemia was initially assessed with blood smear microscopy and retrospectively confirmed using qPCR (**Fig 1B**). Blood-stage infection was detected between 7–11 days after CHMI in mock and non-protected individuals (**Fig 1B**).

Whole blood transcriptome profiles were obtained for samples taken at pre-vaccination baseline (B0), day 0 pre-CHMI (C0), day 1 post-CHMI (C1), and day 7 post-CHMI (C7). High dimensional flow cytometry and single-cell RNA sequencing (scRNA-seq) for immune cell profiling was carried out on PBMC samples obtained via leukapheresis at pre-vaccination baseline (B0), day 6 post-CHMI (C6), and day 112 post-CHMI (C112) (**Fig 1A** and **S1 Table**).

### Single-cell RNA-sequencing elucidates the dynamics of immune cell profiles responding to CHMI

We first investigated responses to *Plasmodium* infection by applying CITE-seq profiling of single cell transcriptomics and selected protein markers to PBMC samples from 4 mock-

**A**

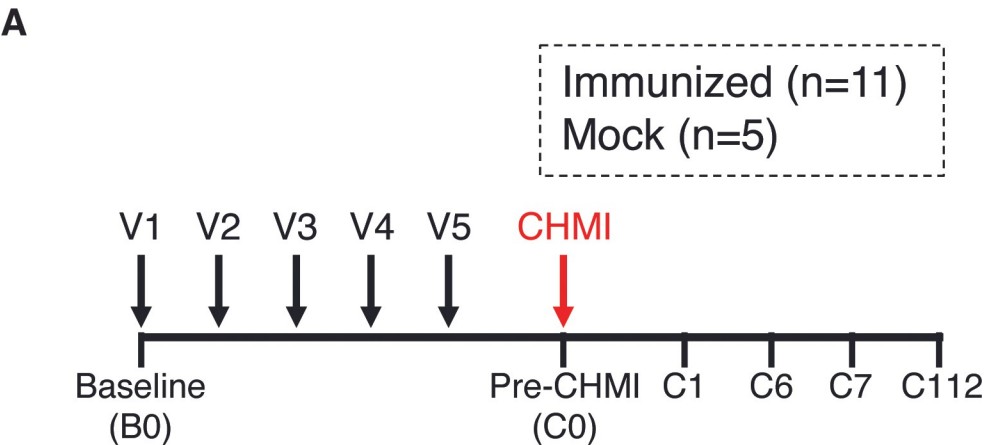

**B**

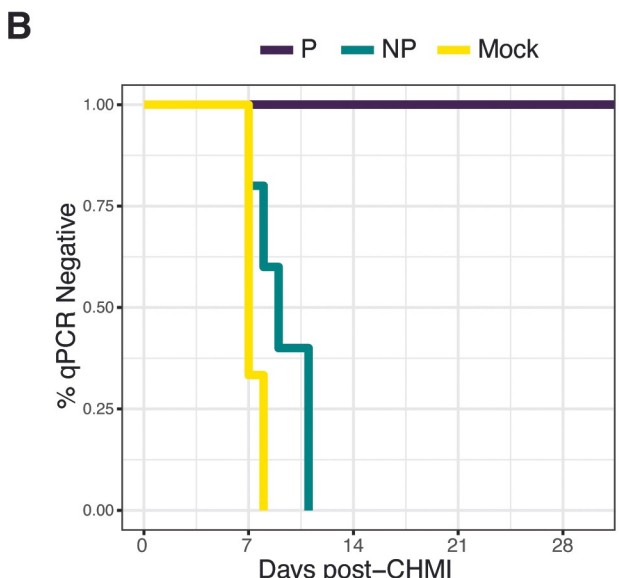

**Fig 1. Immunization by Mosquito bite with RAS (IMRAS) trial. A)** IMRAS trial timeline and sampling time points. Volunteers received 5 vaccine doses (V1-V5), followed by CHMI. The two vaccination groups are shown: Immunized (n = 11) = PfRAS vaccination, mock (n = 5) = non-infected mosquito bites. Time points indicate sampling times used for this study. **B)** Kaplan-Meier analysis showing parasitemia-free survival curves based on qPCR detection of parasites in peripheral blood.

vaccinated individuals at pre-vaccination baseline (B0), and 6 and 112 days (C6, C112) post-CHMI. Unsupervised clustering of the samples revealed a total of 41 distinct cell clusters (**Fig 2A**). Surface protein and gene expression profiles were used to assign cell types to clusters (**S1 Fig** and **S1 Data**). Longitudinal changes in cluster frequency were then calculated using mixed effect linear regression analysis, controlled for within-individual variation. Five clusters significantly changed in frequency over time (FDR<0.01), responding to CHMI (**Fig 2B**). Cluster 20 corresponds to CD4+ Treg with elevated expressions of CD4 and CD25 surface proteins and FoxP3 gene (**Fig 2C**). Cluster 21 was identified as a recently activated CD4+ effector T cell (Teff) with high CD4 and CD5 and low or negative expression of CD25, CD62L, and CD127 surface markers and elevated cell proliferation and growth genes such as RPL4 and EEF1A1

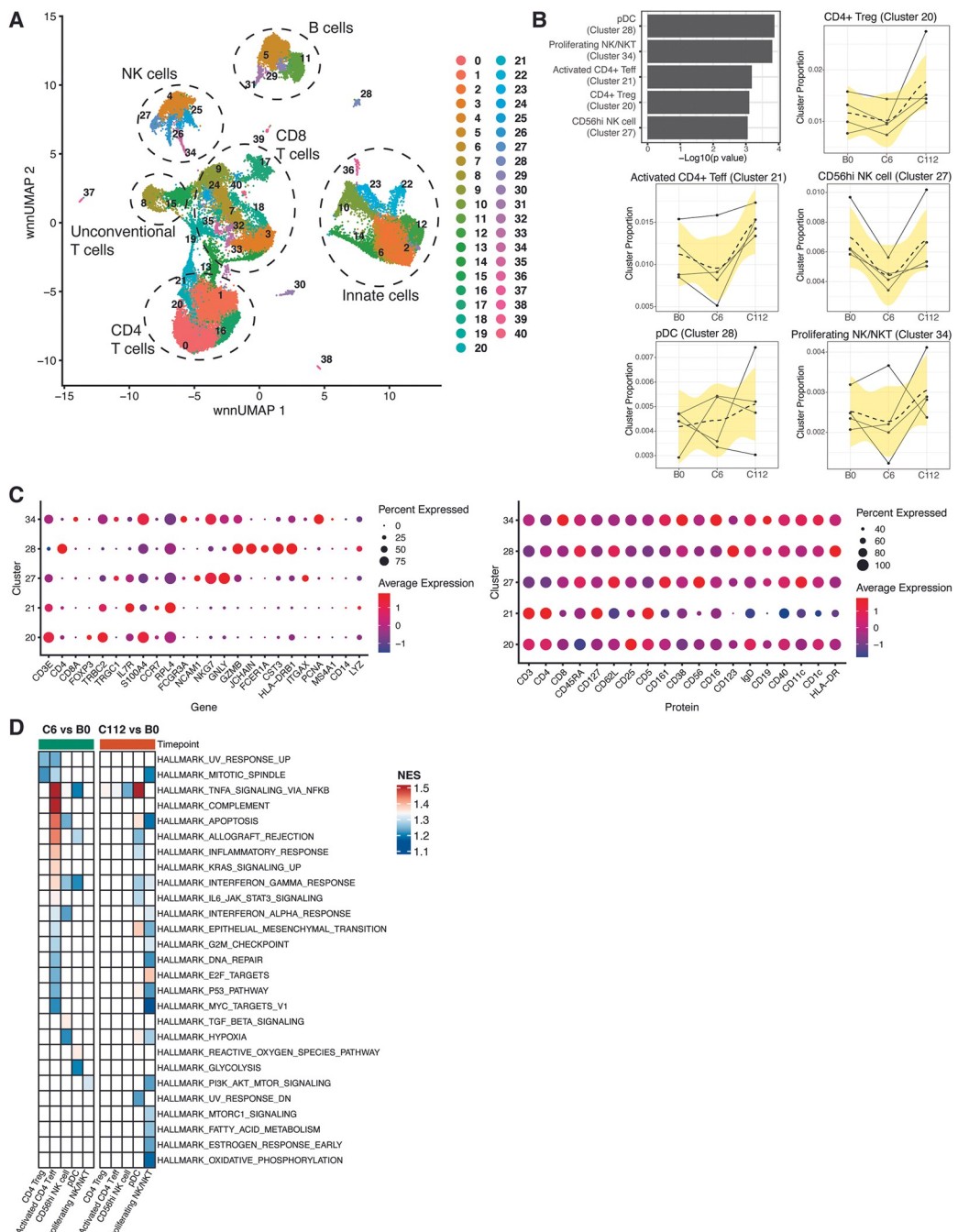

**Fig 2. Immune cell profiling to CHMI in mock-vaccinated individuals using single-cell RNA sequencing. A)** Uniform manifold approximation and projection (UMAP) plot indicating clusters of distinct cell subsets. Numbers indicate cell clusters. High annotation of cell subsets is indicated by the dashed circled lines. **B)** Cell clusters that significantly changed in frequency overtime. Bar plots indicate significantly different cell clusters. Line plots indicate frequency of the significant clusters overtime at baseline pre-vaccination (B0), 6 days post-CHMI (C6), and 112 days post-CHMI (C112). Statistical tests were performed using linear mixed-effect regression with FDR-adjusted P value <0.01. **C)** Gene and protein expression profile for each of the significant clusters. Dot colors indicate average expression level while dot sizes indicate percent of marker expression in the corresponding cluster. **D)** Heatmap plot showing GSEA on comparisons between time points, individually tested in each of the significant clusters. Each square indicates a gene set. Color represents statistically significant gene sets and normalized enrichment score (NES) obtained from GSEA. All gene expression changes were used as inputs for GSEA. Threshold for significant gene sets was set to an FDR-adjusted P value of <0.01. Four (n = 4) mock-vaccinated individuals were selected.

(**Fig 2C**). Both cluster 27 and 34 expressed CD56, and therefore were annotated as NK cells. Proliferating state on cluster 34 and NKT cell annotation were determined with high expression of CD38 and other proliferating genes such as PCNA, PLCLAF, MKI67, and TOP2A, while also expressing the CD8 (**Fig 2C**). Lastly, cluster 28 was identified as plasmacytoid dendritic cell (pDC) with high surface expression of HLA-DR, CD45RA and CD123 accompanied by HLA-DRB1, JCHAIN, and FCER1A gene expression (**Fig 2C**).

A decrease in cluster frequency following CHMI at the C6 time point was observed for all significant clusters with the exception of the pDC cluster, which had an increase in frequency (**Fig 2B**). Frequency of all five significant clusters was higher at 112 days post-CHMI (C112) compared to B0 and C6 (**Fig 2B**), possibly indicating a new baseline immune state post-CHMI or a long-term memory response. We also observed an increased frequency in the γδ T cell cluster (Cluster 15) at C6 and C112 compared to B0 time point, albeit not statistically significant (FDR<0.01). We next performed gene set enrichment analysis (GSEA) [28] on gene expression changes between time points in each of the significant clusters. We determined enrichment of MSigDB Hallmark gene sets comprising sets of coherently expressed genes in blood that represent well-defined biological process [29]. Overall, there was a significant enrichment in gene sets associated with TNFα, IFNα, IFNβ, and inflammatory responses in the cell clusters, most notably in the activated CD4+ Teff at C6 vs B0 time comparison (**Fig 2D**). Both NK cell clusters, CD56hi NK cell (at C6 vs B0) and proliferating NK/NKT cell (at C112 vs B0) were significantly enriched for IFNa and IFNγ responses (**Fig 2D**). Additionally, TNFa and IFNγ gene sets were also significantly enriched in the pDC cluster at both C6 vs B0 and C112 vs B0 time comparisons. Overall, scRNA-seq showed that both innate and adaptive immune cells were responding to CHMI in mock-vaccinated individuals and that these immune cells were characterized with interferon and inflammatory responses.

## Differential gene expressions in response to PfRAS vaccinations and CHMI

We next investigated transcriptomic response to PfRAS vaccinations and CHMI by selecting both PfRAS-vaccinated and mock-vaccinated individuals. Whole blood samples were profiled with RNA sequencing and differentially expressed genes (DEGs) were quantified by examining gene expression changes over time in each vaccination group; protected, non-protected, and mock. DEGs were identified using mixed effect linear regression to account for within-individual variation with an FDR threshold set of 0.2. The pre-CHMI (C0) time point (3 weeks after the fifth RAS vaccination) was compared to the pre-vaccination baseline (B0) time point to determine changes induced by PfRAS vaccination. A total of 52 genes (17 increased, 35 decreased) and 106 genes (15 increased, 91 decreased) were differentially expressed in protected and non-protected vaccinees, respectively (**Fig 3A**). The greatest differential expression at C0 compared to B0 was observed in non-protected vaccinees, possibly indicating immune status change in response to vaccination. 21 genes (11 increased, 10 decreased) were differentially expressed significantly at C0 compared to B0 in the mock group. The transcriptomic changes induced by CHMI compared to C0 were determined at the 1 day (C1) and 7 day (C7) post-challenge time points. Within protected vaccinees, 52 genes were differentially expressed between C1 and C0 while 43 DEGs were observed at C7 relative to C0 (**Fig 2C**). Similarly, a total of 85 and 45 DEGs were observed in non-protected vaccinees at C1 and C7 relative to C0 (**Fig 3A**). In contrast to PfRAS vaccinees, a small number of genes were differentially expressed in mock individuals following CHMI (**Fig 3A**); 12 DEGs and 18 DEGs over C1 and C7 relative to C0.

Further analysis was done to explicitly compare 1) protected versus non-protected vaccinees and 2) mock versus vaccinated individuals at each time point. The greatest number of

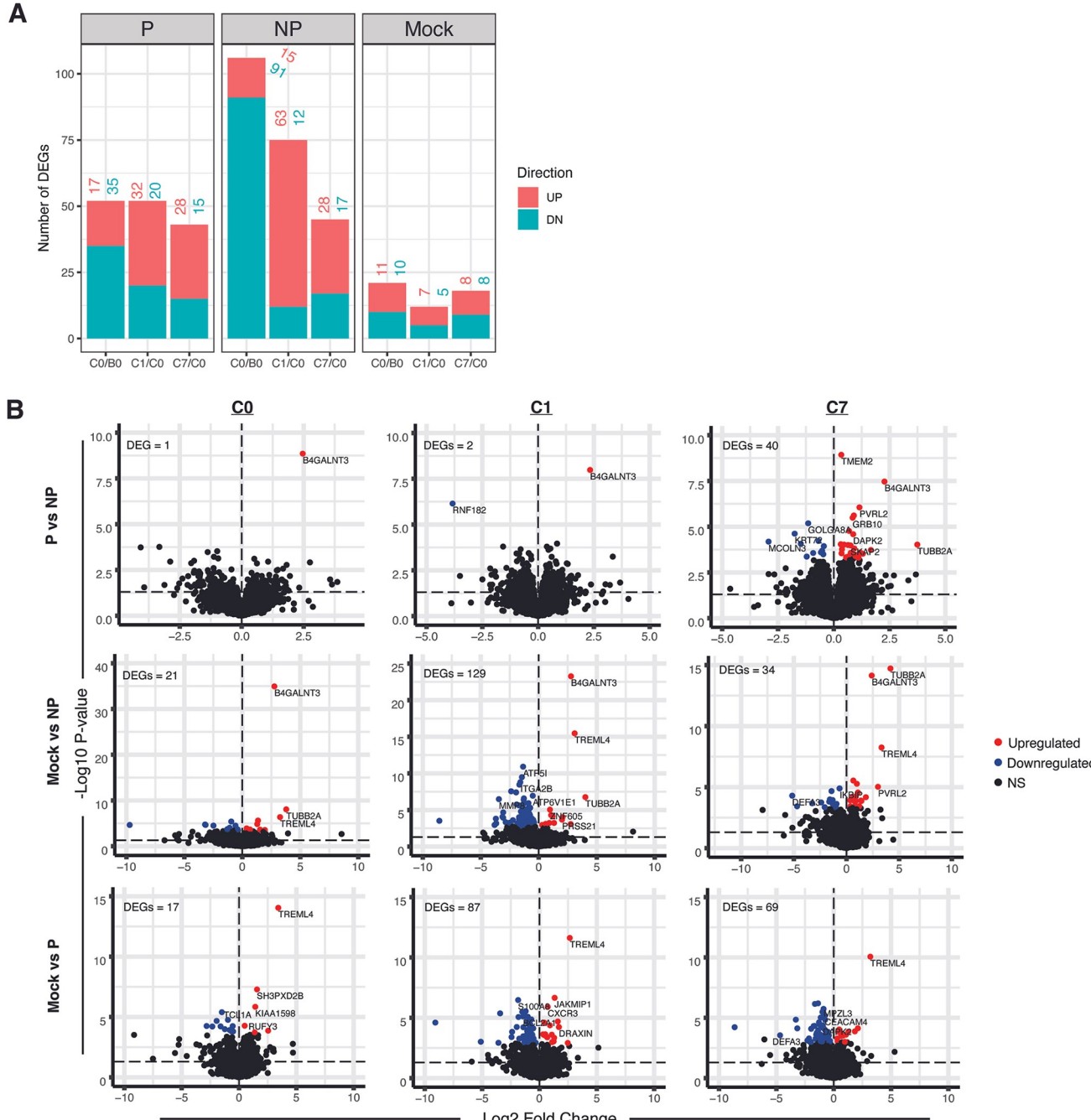

**Fig 3. Gene expression changes to PfRAS vaccination and CHMI. A)** Number of differentially expressed genes (DEGs) calculated using mixed effect linear regression between two time points and for each sample group. The number of DEGs for each group and time point comparisons is indicated above plotted bars. **B)** Volcano plot illustrating DEGs between sample groups at individual time points. Comparisons were determined between P and NP vaccinees and between mock individuals and PfRAS vaccinees, P and NP. Colored dots indicate DEGs after multiple testing correction while black dots indicate non-statistically significant (NS) comparisons. -Log$_{10}$ P values shown on the plots are prior to multiple testing correction. Horizontal line indicates P = 0.05. Threshold for DEGs was set at an FDR-adjusted P value of <0.2. P (n = 6) indicates PfRAS-vaccinated protected individuals, NP (n = 5) indicates PfRAS-vaccinated non-protected individuals, Mock (n = 3) indicates non-infected mosquito bites vaccinated individuals.

DEGs (40 genes) at C7 was observed when comparing protected and non-protected vaccinees (**Fig 3B**). This differential response coincides with the first detection of parasites in peripheral blood in non-protected vaccinees (**Fig 1B**), which likely indicates transcriptomic changes associated with the onset of blood-stage infection. At C0 and C1, only 1 and 2 DEGs were observed between the two PfRAS vaccinee groups, respectively. For comparisons between mock and the PfRAS vaccinee groups, the largest differences were observed at C1 with 129 and 87 DEGs observed on comparisons against non-protected and protected vaccinees, respectively (**Fig 3B**). Notably, more DEGs were seen comparing mock and PfRAS vaccinee groups than between protected and non-protected vaccinees at any time point, suggesting that vaccination was influencing immune response to an infection in non-protected vaccinees. Of note, the genes B4GALNT3 and TUBB2A were consistently reduced in expression in non-protected vaccinees while TREML4 expression was higher in mock individuals compared to the other groups, across all time points, pre- and post-CHMI (**Figs 3B and S2**), suggesting inherent differences at the genetic level that are not induced by vaccination nor CHMI. These genes are involved in protein N-glycosylation (B4GALNT3), intracellular transport (TUBB2A), and modulation of inflammation and immune responses (TREML4).

## Coherently expressed gene sets associated with PfRAS vaccination and protection from malaria

To identify functional responses and biological processes represented by the gene changes observed following PfRAS vaccination and CHMI, we performed an independent GSEA on all gene expression changes using published coherently expressed blood transcriptional modules (BTMs) [30,31] (**S2 Data**). Transcriptional responses to PfRAS vaccination, determined by comparing gene expression at C0 relative to B0, were associated with decreased expression of cell cycle and T cell annotated BTMs, and increased expression of erythrocyte and monocyte BTMs in both protected and non-protected vaccinees (**Figs 4A and S3**). However, increases in interferon and NK-cell-associated responses were observed specifically in protected vaccinees (**Fig 4B**). Interferon signaling has been previously associated with sterile protection after PfRAS vaccination [9]. In non-protected vaccinees, inflammation and monocyte responses were amongst the top upregulated BTMs in the C0 to B0 time interval (**S3 Fig**). In contrast to the PfRAS vaccinees, a lesser number of significantly enriched BTMs were observed in the mock group, with changes in immune responses at C0 compared to B0 associated with increased platelet and erythrocyte-related BTM expression. The observed higher number of significantly enriched BTMs in protected (60 BTMs) and non-protected (86 BTMs) vaccinees compared to mock (33 BTMs) individuals prior to CHMI may indicate transcriptomic changes in response to PfRAS vaccinations. Overlap in the directionality of the BTM response was moderately distributed between sample groups (**Fig 4C**), indicating unique immune response patterns in each group prior to CHMI.

Following CHMI, immune profiles represented by the BTMs were strikingly different between protected and non-protected vaccinees. Between C0 and C1, the protected group showed increases in several interferon-related BTMs and decreases in adaptive immune BTM expression including B and T cells (**Figs 4A and 4B and S3**). In contrast, responses to CHMI in the non-protected group during this time were marked by increases in NK cell-associated BTMs and decreases in BTMs associated with inflammation, neutrophils, and monocytes (**Figs 4A and 4B and S3**). Notably, BTM response profiles in non-protected vaccinees following CHMI seemed to overlap more closely with mock individuals versus protected vaccinees in the C1 vs C0 time interval, with 13 BTMs sharing identical response directionality between non-protected vaccinees and mock individuals, while only 1 BTM response was shared

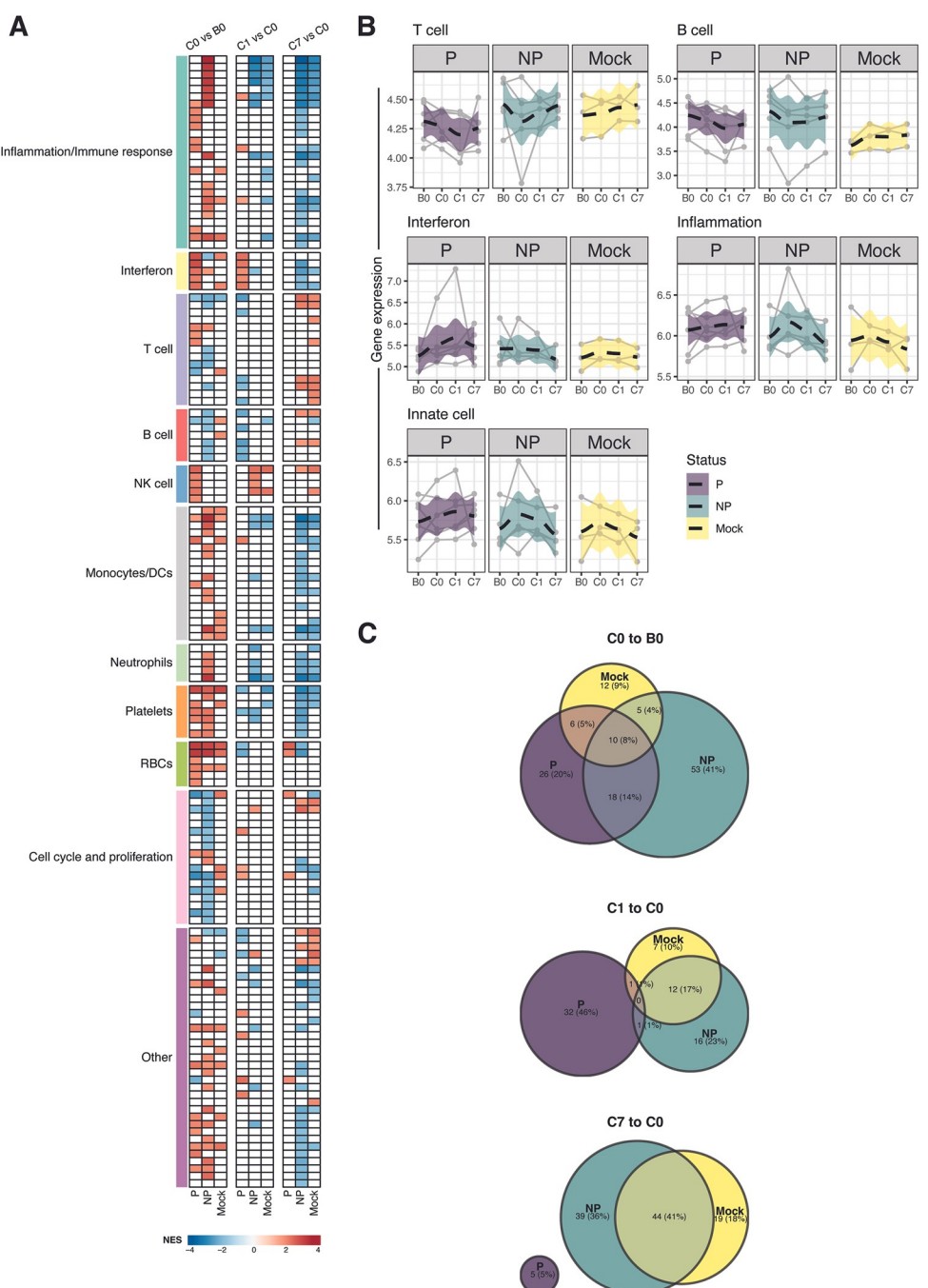

**Fig 4. Blood transcriptional module profile on responses to PfRAS vaccination and CHMI. A)** Gene set enrichment analysis (GSEA) on comparisons between time points, individually tested in each sample group. Each square indicates enrichment of a single of blood transcriptional module (BTM). Colored row annotations represent high-level annotations of the BTMs. Square color represents statistically significant gene sets and normalized enrichment score (NES) obtained from GSEA. All gene expression changes were used as inputs for GSEA. Threshold for significant gene sets was set at an FDR-adjusted P value of <0.01. **B)** Average gene expression of representative high-level BTM annotation. Lines indicate each individual. Dashed lines indicate locally estimated scatterplot smoothing (LOESS) regression with 95% confidence interval shown in the highlighted color. Transcript abundance is in log-transformed counts per million (logCPM). **C)** Venn diagrams showing proportions of shared significant BTMs, with identical NES direction, between sample groups at each time interval. P (n = 6) indicates PfRAS-vaccinated protected individuals, NP (n = 5) indicates PfRAS-vaccinated non-protected individuals, Mock (n = 3) indicates non-infected mosquito bites vaccinated individuals.

between non-protected and protected vaccinees (**Fig 4C**). This similarity in BTM responses following CHMI between non-protected vaccinees and mock individuals continued at time point C7, when blood-stage infection was detected and with shared BTMs representing increased expression in T cells, NK cells, cell cycle, and other biological processes and decreased expression in inflammation, interferon, neutrophils, and monocytes (**Fig 4A**). These immune responses were consistent with our scRNA-seq data, where cell clusters that were reduced in frequency at C6 also primarily mediate interferon and inflammatory responses (**Fig 2B**). Only a few BTMs were significantly enriched at C7 compared to C0 in protected vaccinees, and none of these BTMs shared responses of similar directionality to non-protected vaccinees or mock individuals (**Fig 4C**). These included erythrocyte and cell cycle associated BTMs, perhaps indicating a return to the pre-CHMI state following parasite clearance in protected vaccinees. Additional GSEA using the Hallmark gene sets revealed that both type I (IFNα) and type II (IFNγ) were increased in protected vaccinees between C1 and C0 time point while gene sets associated with inflammation were decreased in non-protected vaccinees and mock individuals at C7 relative to C0 (**S4 Fig**).

To identify protection-associated genes, we selected leading-edge genes that were present in more than half of the significant gene sets from the interferon and NK cell-associated BTMs in protected vaccinees. A total of 5 and 3 genes were present in more than half of the significant interferon BTMs at C0 vs B0 and C1 vs C0 time comparison, respectively (S5A Fig). These genes include HERC5, IFIH1, IFIT1, IRF7, and RSAD2 and were all associated with interferon response. Consistently, expression level of the interferon genes was elevated following PfRAS vaccination and peaked at 1 day following CHMI in the protected vaccinees (S5B Fig). NK cell-associated BTMs were significant only at C0 vs B0 time comparison in protected vaccinees, of which IL2RB and TGFBR2 gene were present in 4 of the 5 significant BTMs (S5A Fig). Eight other genes including FASLG, GPR56, KLRB1, NKG7, S1PR5, and the KIR gene family (KIR2DL1, KIR2DL3, and KIR3DL1), were present in 3 significant BTMs. IL2RB and KIR gene expression levels peaked at C0 in protected vaccinees (S5B Fig). Interestingly, expression level of the regulatory gene, TGFBR2, was reduced at C0 and peaked at C1 in protected vaccinees.

## Changes in biological pathways and deconvoluted cell responses following PfRAS vaccination and CHMI

To further understand the pathways that lead to immune processes observed in the GSEA, we selected leading-edge genes from the significantly enriched BTMs and functionally profiled them using Ingenuity Pathway Analysis (IPA) [32]. Comparing changes over the full course of vaccinations, i.e. between B0 and C0, IPA predicted activation in pathways related to interferon signaling and NK cells in protected vaccinees (**Fig 5A** **and S3 Data**). By comparison, pathways associated with innate cells, including phagocytes and monocytes were activated in non-protected vaccinees, and EIF2 signaling, a pathway related to protein synthesis, was activated in mock individuals in the same time interval. One day following CHMI, at C1 relative to C0, pathways associated with protein synthesis (EIF2) and mitochondrial activity (mTOR) were significantly activated in protected vaccinees. In non-protected and mock groups, innate and phagocyte related pathways, including phagosome formation, NK cell signalling, Fcγ receptor-mediated phagocytosis in macrophages and monocytes, and production of nitric oxide and reactive oxygen species in macrophages were inactivated between C1 and C0 as well as C7 and C0 time intervals. Curiously, NK cell-related modules which were upregulated in non-protected vaccinees in GSEA (**Figs 4A and S3**), were inactivated in IPA at C1 vs C0 (NK cell signaling) (**Fig 5A** **and S3 Data**). At C7 relative to C0, pathways related to erythropoiesis

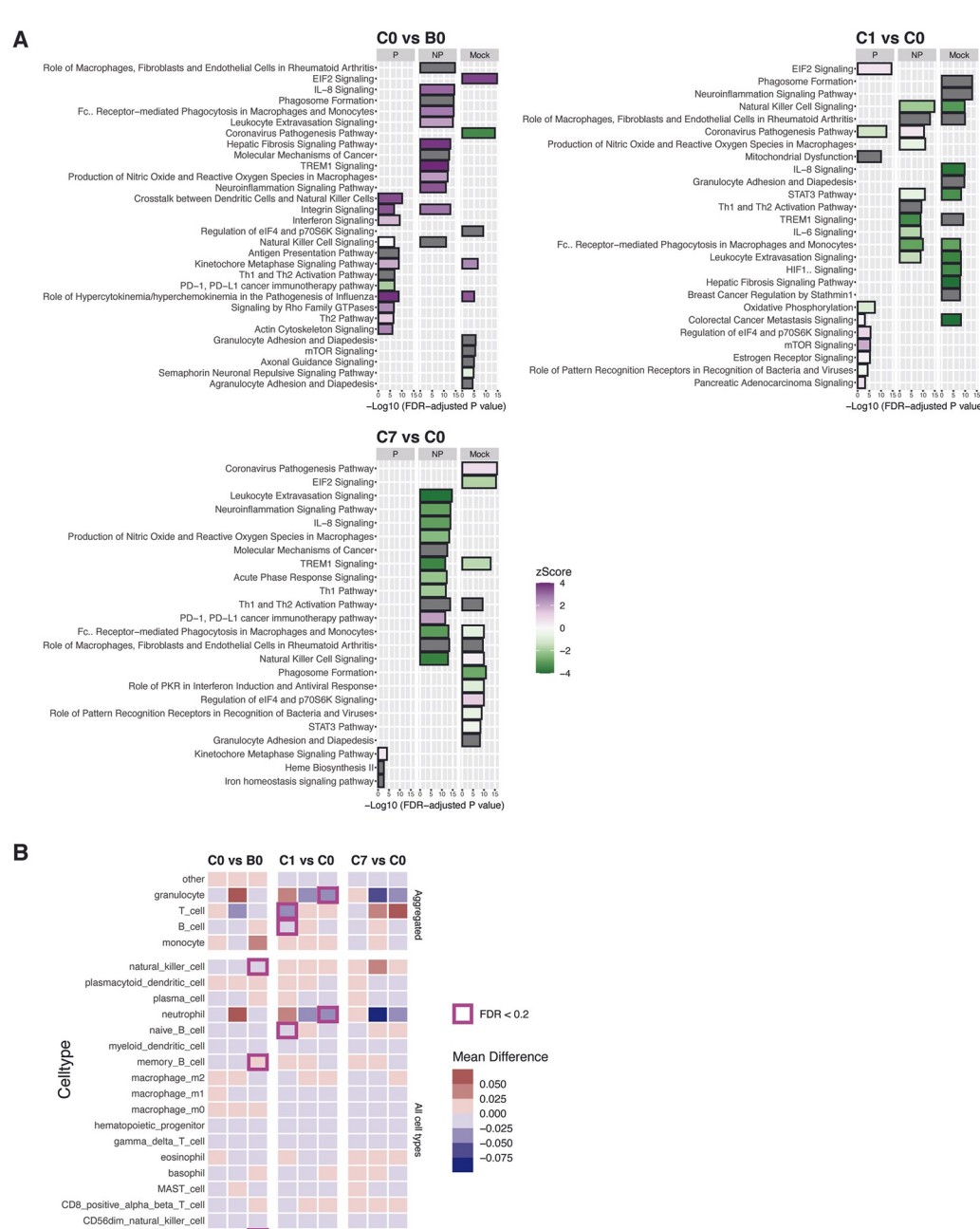

**Fig 5. Biological pathway and transcriptome cell subset response. A)** Ingenuity pathway analysis (IPA) on leading edge genes from significantly enriched gene sets. Leading edge genes were obtained from significantly enriched gene set modules in **Fig 3A**. Top 10 most significantly predicted pathways are shown in each group and time point. Bar width indicates the strength of IPA prediction following FDR P value adjustment. Colors indicate zScore prediction state; activated (purple), inactivated (green), and undetermined (grey) or z-score = 0 (white). **B)** ImmunoStates cell deconvolution analysis on comparisons between time points, individually tested in each sample group. Each square indicates a cell type. Color represents mean difference in cell proportion between time points. Squares outlined in purple represent significant statistical comparisons of FDR-adjusted P value <0.2. P (n = 6) indicates PfRAS-vaccinated protected individuals, NP (n = 5) indicates PfRAS-vaccinated non-protected individuals, Mock (n = 3) indicates non-infected mosquito bites vaccinated individuals.

and cell cycle were activated in protected vaccinees. Overall, both IPA and GSEA showed concordant results that indicate immune responses associated with interferon in protected vaccinees and innate responses associated with non-protected vaccinees and mock individuals.

To determine the cell types that contribute to the bulk transcriptomic responses, we performed cell-mixture deconvolution using a previously described reference expression matrix, immunoStates [33]. Consistent with GSEA, significant reductions in cell proportion were observed for the aggregate T and B cell response in protected vaccinees while significant reductions in proportion of granulocytes and neutrophils were also indicated in mock individuals after CHMI at C1 vs C0 (**Fig 5B**).

## Distinct dynamics of immune cell subsets between immune status group following CHMI

To quantify changes in immune cell frequency after CHMI, we performed a comprehensive immunophenotyping by high-dimensional flow cytometry with 4 different antibody panels on PBMC samples collected at pre-vaccination baseline (B0) and 6 days (C6) and 112 days (C112) after CHMI. Unsupervised clustering using Seurat [34] identified a total of 50 clusters with distinct marker expression profiles (**S5 Fig**). Information on antibody marker expressions was used to annotate the clusters (**S5 Fig** and **S4 Data**). Mixed effect linear regression analysis was used to calculate changes in cluster frequency over time and to identify statistically significant differences between groups. We identified 3 clusters that significantly changed in frequency overtime and were significantly different between protected and non-protected vaccinees (**Fig 6A**). Cluster 6 of the invariant T cell panel expressed TCRγδ and vδ2, indicating a vδ2+ γδ T cell subset (**Figs S5** and **S4 Data**). Cluster 3 of the DC panel had high expression of CD16, HLA-DR, CD11c, and CX3CR1 and low expression of CD14, suggesting that this is a non-classical monocyte subset. CD8 and CD56 surface markers were highly expressed on cluster 4 of the T cell panel while CD45RA and CCR7 marker were absent, and therefore, this cluster was designated as CD56+ CD8+ T effector memory cells (Tem). Consistent with other studies that reported an increase in γδ T cell frequency following CHMI [35,36], vδ2+ γδ T cell cluster frequency was higher at C6 compared to B0 in all groups (**Fig 6B**). Interestingly, the frequency of the vδ2+ γδ T cell cluster was increased at C112 relative to C6 in non-protected vaccinees and mock individuals, long after blood-stage parasitemia had resolved. Of note, we did not observe significant changes in the frequency of non- vδ2+ γδ T cell (Cluster 9 of the invariant T cell panel) following PfRAS vaccination and CHMI, as reported in other studies [37]. The decrease in non-classical monocyte cluster frequency in non-protected vaccinees and mock individuals at C6 was consistent with our GSEA results, which showed a decreased expression of gene sets associated with innate cells at C7 compared to C0 (**Figs 6B and 4A and S3**), albeit the sampling time points differed by a day between the two datasets. Similar to the vδ2+ γδ T cell cluster, the CD8+ Tem cluster expressing CD56 surface protein was also expanded over time in non-protected vaccinees and mock individuals after CHMI (**Fig 6B**), indicating the effect of exposures to blood-stage parasites.

To better understand the effect of PfRAS vaccinations on immune cell development and immune response to CHMI, comparisons were made between mock individuals and protected and non-protected vaccinees. A total of 4 and 7 clusters were significantly different between mock individuals and non-protected vaccinees and between mock individuals and protected vaccinees, respectively. These clusters represent CD56+ CD8+ Tems (cluster 4 –T cell panel), CD8+ T cells expressing CD57+ (cluster 4 –invariant t cell panel), atypical memory B cells (MBCs) (cluster 6 –B cell panel), naïve B cells expressing FCRL5+ (cluster 5 –B cell panel), non-classical monocytes (cluster 3 –DC panel), double negative (DN) CD3+ T cells (cluster 12

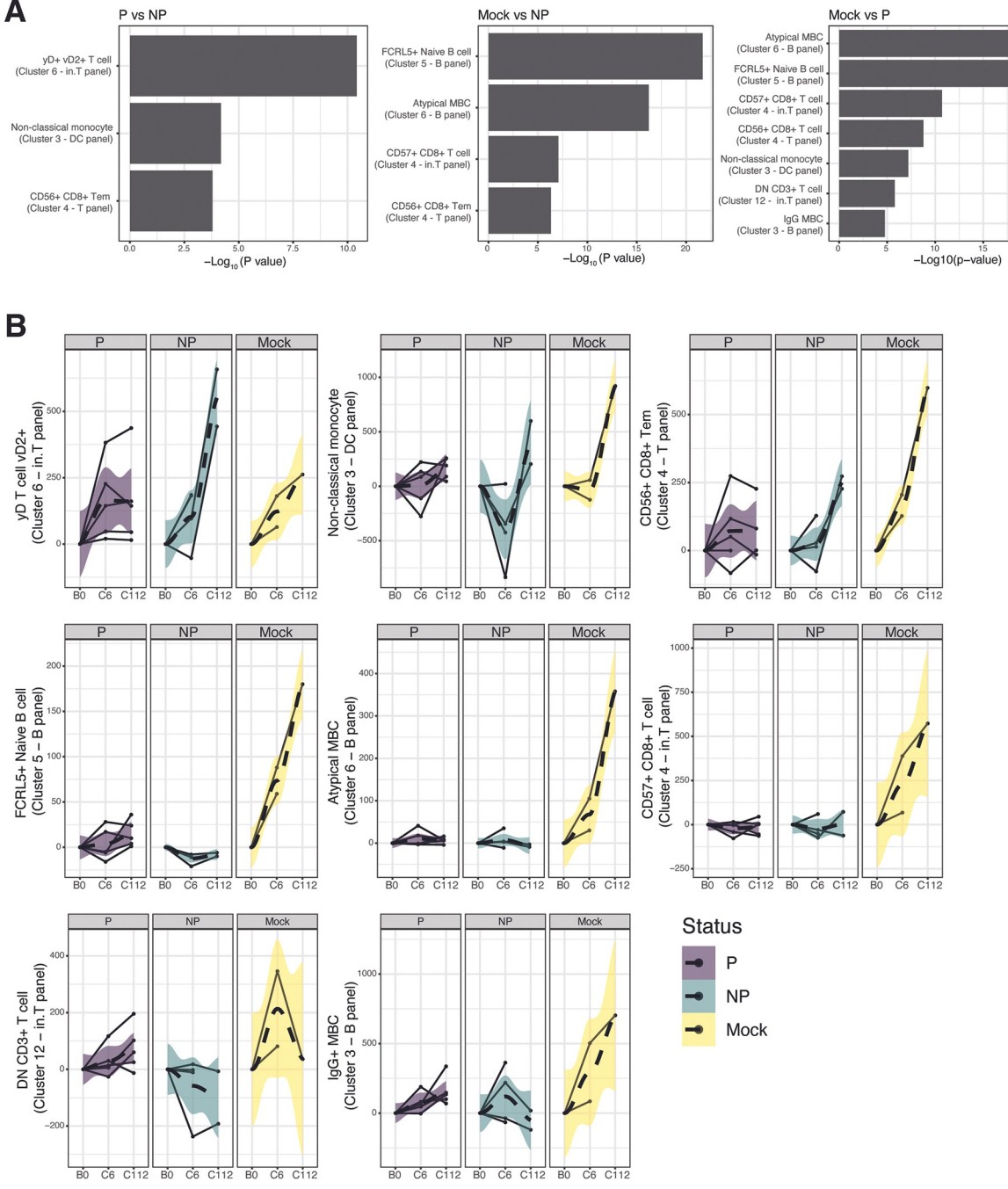

**Fig 6. Distinct cell subset responses following PfRAS and CHMI.** Cell subsets were identified using unsupervised clustering cells based on normalized marker fluorescence intensities, and annotated as described in **S6 Fig** and **S4 Data**. **A)** Bar plot showing significantly different cell subsets comparing protected vs. non-protected groups and mock vs. protected and non-protected groups. Significant clusters were identified using FDR adjusted P-values derived from linear mixed-effect model where FDR<0.05. **B)** Line plots showing significant cluster frequencies, calculated relative to pre-vaccination baseline (B0), for protected, non-protected, and mock individuals. Each solid black line represents an individual while dashed lines represent locally estimated scatterplot smoothing (LOESS) regression for the group, with 95% confidence interval indicated by the shaded areas. P (n = 6) indicates PfRAS-vaccinated protected individuals, NP (n = 5) indicates PfRAS-vaccinated non-protected individuals, Mock (n = 3) indicates non-infected mosquito bites vaccinated individuals.

–invariant T cell panel), and IgG MBCs (cluster 3 –B cell panel) (**Fig 6A**). The increase in the frequency of the CD56+ CD8+ Tem cluster over the course of the trial was significantly higher in the mock group compared to PfRAS vaccinated groups (**Fig 6B**), possibly indicating a response to higher parasitemia in the mock group consistent with parasite qPCR data (**Fig 1B**). A similar increase in frequency in mock-vaccinated individuals was also observed for a CD8+ T cell cluster expressing CD57, suggesting a senescent and highly cytotoxic CD8 T cell. Curiously, the FCRL5 marker that was previously associated with impaired B cell responses in malaria [38], was highly expressed in a naïve B cell and atypical MBC cluster, and in a higher frequency in mock individuals compared to PfRAS vaccinated groups (**Fig 6B**). Clusters that were exclusively different between mock individuals and protected vaccinees, but not non-protected vaccinees included non-classical monocytes, DN CD3+ T cells, and IgG MBCs. There was a similar reduction in frequency in the non-classical monocyte cluster in mock individuals and non-protected vaccinees at C6 relative to B0 whereas DN CD3+ T cell and IgG MBCs had a larger increase in frequency following CHMI in mock individuals compared to protected vaccinees (**Fig 6B**). Of note, our observations in cell phenotyping data from flow cytometry were consistent with the scRNA-seq data, where we observed an overall increase in frequency across many cell types at convalescent C112 time point. Both flow cytometry and scRNA-seq also identified a cell cluster that was associated with CD56+ CD8+ T cell, i.e. cluster Cluster 4 (T panel–flow cytometry) and Cluster 34 (scRNA-seq) (**Fig 6B**). Overall, cell phenotyping data largely support transcriptomic observations of distinct immune responses among study individuals.

## Discussion

In this study, we observed that systemic immune responses to *P. falciparum* infection are distinct between protected and non-protected PfRAS vaccinees as well as mock-vaccinated individuals. Blood transcriptome profiles in the PfRAS vaccinees diverged following CHMI in which non-protected vaccinees and mock individuals shared a more similar profile than protected vaccinees, before and after they developed blood-stage parasitemia. Transcriptomic responses in protected vaccinees were associated with early increase in NK cell and interferon (type I and II) responses and decreased expression of adaptive immunity, including T and B cells. In contrast, non-protected vaccinees and mock individuals had transcriptomic responses associated with decreased inflammation and innate cell-related signatures such as neutrophils and monocytes. Data from immunophenotyping revealed that PfRAS vaccinations and CHMI were associated with an increased frequency of vδ2+ γδ T cells in peripheral blood which remained elevated at convalescence, regardless of protection status. Non-classical monocytes were reduced in mock individuals and non-protected vaccinees but not protected vaccinees following CHMI, consistent with the whole blood transcriptomic response. Immune signatures related to malarial infection from CHMI in mock individuals were characterized by higher expression of the senescence and cytotoxic CD57 protein on CD8+ T cells and intriguingly, by higher expression of the dysfunctional marker FCRL5 on atypical MBCs and naïve B cells. Our study has revealed important differences in immunological response patterns among protected and non-protected PfRAS vaccinees and mock-vaccinated individuals following malarial infection providing additional insights into immune mechanisms required to achieve sterilizing immunity.

Interferon responses, including type I IFN and IFNγ, have been associated with the suppression of liver-stage infection in mice [39]. Conversely, we found that both type I and type II interferon responses were largely unique to protected PfRAS vaccinees, prior to and one day post CHMI. Such responses were largely absent in both non-protected vaccinees and mock

individuals. Other clinical studies analyzing transcriptomic profiles of malaria-specific PBMCs have also shown increases of interferon responses in protected compared to non-protected individuals vaccinated with RTS,S or CPS vaccine [24]. Nonetheless, type I IFN responses have also been associated with reduced protective efficacy via impairment of CD8+ T cell responses in GAP-immunized mice [40]. It is likely that optimal interferon responses are required to induce sterilizing immunity, and that insufficient or excessive response could lead to a lack of protection. Our previous studies have demonstrated that the dynamics of transcriptome responses were crucial in determining protection outcome, in which strong and early increased interferon and inflammatory-associated responses following the first dose of PfRAS vaccination were associated with lack of protection [41–43]. Although it is unclear which cell types are mediating interferon responses in protected vaccinees, our scRNA-seq data suggest that in mock-vaccinated individuals, T and NK cells were the primary cell subsets that responded to CHMI and induced interferons.

In addition to IFN responses, we observed early decreased expression in T and B cell-related BTMs in protected vaccinees one day after CHMI. Given that such observations are lacking in the non-protected and mock-vaccinated individuals, we speculate that these adaptive immune modules represent immune cells that responded to parasitic infection and trafficked into the liver to target sporozoites and activate effector functions. Different subsets of CD8+ T cells can contribute to sterilizing immunity in malaria, including the memory and liver resident subsets [17,44,45]. These CD8+ T cells are thought to mediate anti-parasitic effector mechanisms through IFNγ secretion [9,18,46–48]. Although it is widely perceived that non-circulating liver resident CD8+ T cells are the main subset targeting liver stage malaria, recent work has demonstrated that circulating antigen-specific CD8+ Tem could rapidly infiltrate liver cells during infection and mediate parasite clearance [49]. It is possible that our observation of decreased expression of T cell-related BTMs in the blood in protected vaccinees were indicative of parasite-specific CD8+ Tems that may also secrete IFNγ. Along with T cells, B cells could also mediate sporozoite clearance by generating antibodies that function through multiple mechanisms, including complement-fixation [50] and interactions with Fcγ receptors on various immune cells [51]. Nonetheless, other immune cells are also likely to contribute to the development of sterilizing immunity to malaria, including the γδ T cells [20,21], CD4+ T cells [16], DCs [23,52], and NK cells [22]. In contrast to protected vaccinees, an increase in BTMs associated with the adaptive immune response was only observed 7 days post CHMI in non-protected vaccinees and mock individuals and correspond with blood-stage parasitemia, suggesting the expansion of adaptive immune cells in response to malarial infection and parasite replication.

Neutrophils and monocytes are innate immune cells that provide the first line of defense against infection. Early activation of these cells during malarial infection have been described in CHMI and cross-sectional malaria studies [53–55]. Unexpectedly, while most previous studies in *P. falciparum* and *P. vivax* reported an increase in innate cell frequency and activated inflammatory pathways after malarial infection [53–58], we observed decreased expression in innate-cell associated BTMs and reduced frequency of non-classical monocytes specifically in non-protected and mock groups after CHMI. BTMs associated with inflammation were also decreased in these CHMI-susceptible groups. Different sampling times post-CHMI or after natural infection and past malaria exposures could explain the discrepancies in the direction of innate cell frequency or enrichment score between studies. It is possible that the decreased innate cell response observed in our cohort indicates trafficking activity of these cells into the tissue or liver where sporozoite infection occurred and that these innate cells were also responsible for mediating inflammatory responses. Data from mice showed that neutrophils and monocytes can rapidly expand in the skin within hours after intradermal injection of

irradiated sporozoites [59], demonstrating rapid recruitment of these cells to the sites of infection in the tissue. Other clinical studies have also reported transcriptomic activation in inflammatory responses post-CHMI in malaria naïve individuals, albeit with an opposite enrichment score direction to our study [58]. Thus, we hypothesize that the non-protected and mock groups in our cohort lacked sufficient parasite-specific adaptive immune cells that target and suppress parasite replication after CHMI, which subsequently resulted in the activation of innate cells in order to further capture and present parasite antigens to the adaptive immune cells. In line with this hypothesis, BTMs associated with adaptive immune cells were significantly enriched while innate cells were largely absent in the protected group after CHMI.

While we also observed increases in NK cell-related BTMs in non-protected vaccinees and mock individuals after CHMI, the same BTMs were also increased in expression earlier prior to CHMI in protected vaccinees (**S3 Fig**), suggesting that early NK cell response may be pivotal to protection outcomes against malaria. It is unknown if these NK cells were specific to *Plasmodium* parasites or if they indicate the evidence of adaptive NK cells via trained immunity. Studies in mice have demonstrated that hepatic NK cells can acquire memory-like and antigen-specific capacity to viral antigens [60]. In malaria, the protective capacity of the CD8 T cell response is dependent on the NK cell response [22]. Our results showed that several of the KIR genes including KIR2DL1, KIR2DL3, and KIR3DL1, were directing the overall NK cell BTM response in the protected vaccinees prior to CHMI. These KIR genes are classified as inhibitory KIRs, due to long intracytoplasmic tails [61]. Coincidentally, CD56$^{dim}$ NK cells express diverse inhibitory KIRs for HLA class I and can inhibit *P. falciparum* growth through antibody-dependent cellular cytotoxicity (ADCC) [62]. It is possible that CD56$^{dim}$ NK subset was mediating malaria immunity in protected vaccinees in our study, albeit it is unclear if this was induced by PfRAS vaccination or due to genetic variation of the highly polymorphic KIR genes between individuals [63]. An *in-vitro* study exposing cells from different malaria-naïve donors with parasitized RBCs showed that NK cell activation profile was heterogenous between individuals [64]. Thus, the early and elevated NK cell response in protected vaccinees in this study could also be explained by genetic variation between individuals.

Immunophenotyping by flow cytometry revealed an increase in vδ2+ γδ T cells in all sample groups following CHMI. This increase was sustained out to 112 days post-CHMI in non-protected vaccinees and mock individuals and was higher compared to protected vaccinees at convalescence. Similar longitudinal expansion in vδ2+ γδ T cells in mock individuals was also observed from the scRNA-seq data, although this was not statistically significant. Expansion of γδ T cells in subsets expressing vγ9+ vδ2+ chain, has been consistently reported in different malaria settings, including in vaccine trials [18,20], CHMI [37,65,66], and natural infection [37,67,68]. Late expansion of vδ2+ γδ T cells following acute parasitemia has also been previously reported [37,69,70], up to 35 days post-CHMI. While vδ2+ γδ T cells is often considered as innate-like, responding to phosphoantigens (Pags) [71,72], we observed long-lived and late expansion of vδ2+ γδ T cells 112 days after CHMI and after parasite clearance. A similar observation was also reported in macaques, with expansion of vδ2+ γδ T cells that persisted up to 7 months after a second BCG vaccination [73]. While data from mice and humans suggest that γδ T cells play a role in mediating malaria protection [20,21], our results suggest that vδ2+ γδ T cells expand along with parasite exposures, from either or both sporozoites and infected RBCs. This expansion is also likely mediated from a TCR-mediated clonal expansion, as previously demonstrated in mice [69]. Contrary to a previous study that saw an increase in vδ1+ γδ T cell frequency and its differentiation into effector subsets after CHMI [37], we did not observe any significant changes in the non-vδ2+ γδ T cell fraction. This could possibly be due to lack of vδ1 marker in our study or due to differences in study design, that is, PfRAS vaccination followed by CHMI versus repeated CHMI exposures. Specific to our study and PfRAS vaccination

strategy, our results suggest that the vδ2+ γδ T cells did not play a primary role in mediating protection and that sterile protection was mainly mediated by the adaptive immune cells.

When comparing immunophenotyping profile between PfRAS vaccinees (both protected and non-protected) and mock individuals, we observed a higher frequency of cell subsets expressing the senescent marker CD57 on CD8 T cells and the dysfunction marker FCRL5 on atypical MBCs and naïve B cells in the mock group. CD57 expression on T cells has been implicated in reduced malaria immunity, in which the frequency of CD57+ T cells was higher in symptomatic malaria individuals compared to asymptomatic and healthy individuals [74]. Additionally, FCRL5+ atypical MBCs with reduced function have been associated with increased exposure to malaria [38]. Increased frequency in these cell subsets may indicate dysfunctional cell responses due to over-replication of parasitemia.

Our study has several limitations. First, we were unable to directly test for correlations between gene expression and cell type profiling data due to a lack of matching CHMI time points between blood and PBMC samples. Further, the relatively low sample size and missing sampling time points in the PBMC data are limitations that reduce the statistical power and sensitivity of the study. We utilized linear mixed-effect regression on the overall time points to min this limitation and to investigate the overall change in transcriptome cell type responses in the cohort. Using this approach, we were nonetheless able to identify genes and cell subsets that responded to PfRAS and CHMI and were associated with malaria protection. Inherent differences between study volunteers could also contribute to differences in immune response to PfRAS vaccination and CHMI as well as to protection outcomes in our study. Indeed, elevated inflammatory responses states prior to any immunization have been associated with malaria protection [75]. Given that volunteers in our study were given a suboptimal dose, it may be that higher vaccination dose is required to overcome these variations and induce a higher protection rate. Although our data have indicated that early transcriptome interferon responses post-CHMI are associated with protection, we could not demonstrate if the mRNA transcripts are translated into protein synthesis and secretion in the blood. Lastly, it is important to note that the immune responses described in our cohort do not reflect antigen-specific nor tissue-specific immune responses, rather, they reveal total responses in the peripheral blood. Further studies may employ techniques that can characterize antigen-specific responses such as in-vitro stimulation or antigen-coupled tetramers as well as comparing tissue-specific to circulating immune responses.

## Conclusion

Our data revealed different trajectories of transcriptome and cell type immune signatures between protected and non-protected PfRAS vaccinees and mock-vaccinated individuals following CHMI. Immune responses among those protected from *P. falciparum* infection exhibited early and rapid immune responses associated with type I and II interferon, NK cells, and adaptive immune cells while responses in non-protected individuals, either PfRAS-vaccinated or mock-vaccinated, exhibited sustained but distinct responses associated with innate immune cell and inflammation. Our findings provide additional insights into understanding the mechanisms underlying immune responses to malarial vaccination and infection. Further studies are warranted to confirm our findings in a larger cohort size and in characterizing parasite-specific cellular immune response and function that mediate protection from malaria.

## Materials and methods

### Ethics statement

Written informed consent was obtained from all study participants. Ethics approval was obtained from the Naval Medical Research Center (NMRC) Institutional Review Board. This

study was conducted according to the Declaration Helsinki and Good Clinical Practices under the guidelines of United States Food and Drug Administration Investigational New Drug (IND) application BB-15767. The trial was registered on ClinicalTrials.gov (NCT01994525).

## Study design

Details of the open-label, single center, and randomized clinical trial, termed IMRAS (Immunization by Mosquito bite with Radiation-attenuated Sporozoite) have been described previously (Hickey, 2020). Briefly, healthy, malaria naïve, and non-pregnant individuals aged 18–35 years old with no previous history of malaria were included in this study. Subjects were randomized and received five vaccination doses of approximately 200 bites per dose from *Anopheles stephensi* mosquitoes either infected with irradiated *P. falciparum* (PfRAS) NF54 or non-infected. The first four doses were administered at 4-weeks interval followed by the fifth dose after 5 weeks. Malaria protection was assessed with controlled human malaria infection (CHMI) using homologous wild-type *P. falciparum* sporozoite administered by mosquito bites at 3 weeks following vaccinations.

## Sample collection

Peripheral whole blood was collected from trial subjects directly into PAXgene blood RNA tubes (PreAnalytiX) and stored in -80˚C. The following time points were used for this study; pre-vaccination baseline (B0), pre-CHMI (C0), 1 and 7 days post-CHMI (C1 and C7, respectively).

Peripheral blood mononuclear cells (PBMCs) were collected from leukapheresis procedure and stored in liquid nitrogen for later use. The following time points were used for this study; pre-vaccination baseline (B0), 5/6 and 112 days post-CHMI (C6 and C112, respectively).

## RNA sequencing and data generation

Total RNA was extracted from whole blood using PAXgene Blood RNA Kit (PreAnalytiX), followed by treatment with GLOBINclear™ kit (ThermoFisher Scientific) to remove unwanted globin mRNA. The remaining RNA product was used to prepare cDNA library using Illumina TruSeq Stranded mRNA library preparation kit (Illumina). RNAseq was performed by Beijing Genomics Institute using the Illumina Hiseq2000 with 75 base-pair (bp) paired-end reads to a depth of at least 30 million reads per sample.

RNAseq output was processed as previously described [76]. Read pairs were pre-processed to adjust base calls with phred <5 to 'N' and to remove read pairs where either end had fewer than 30 unambiguous base calls. This method indirectly removes read pairs containing mostly adaptor sequences. Read pairs were aligned to the human genome (hg19) using STAR (v2.3.1d) [77]. Gene counts were tabulated using htseq (v0.6.0) with the intersection-strict setting turned on and Ensembl gene annotations (GRCh37.74) used to map genomic locations to gene identifiers [78]. The edgeR (v3.36.0) package function, cpm, was used to calculate TMM-normalized counts-per-million (CPM) expression matrices [79].

## Immunophenotyping using flow cytometry

PBMCs were thawed in RPMI 1640 (Gibco) supplemented with 10% fetal bovine serum (FBS) and 0.05 U/mL benzonase nuclease (Millipore). The cells were initially stained and incubated with LIVE/DEAD™ Fixable Blue Dead Cell Stain Kit (ThermoFisher) and Human FC Block (BD Bioscience) for 30 minutes at room temperature, before subsequently stained with four different phenotyping panels that were previously described [80,81] (**S2 Table**). The cells were

then acquired on BD FACSymphony flow cytometer. Cell population gating and data pre-processing for subsequent clustering analyses were done in FlowJo (v.9.6.6).

## Single cell RNA sequencing and CITE-seq

PBMC samples from mock individuals (n = 4) in cohort 1 (subjects S049 and S054) and 2 (subjects S104 and S137) of the IMRAS trial were obtained at day 0 (B0), 6 days post-CHMI (C6), and 112 days post-CHMI (C112). Cells were thawed and counted and processed over three batches in combination with other samples entering the same processing pipeline (batch 1 = S049 all timepoints; batch 2 = S054 and S104 all timepoints; batch 3 = S137 all timepoints). $5 \times 10^5$ cells from each sample were stained for combinatorial hashing using oligonucleotide-conjugated Hashtag antibodies targeting the ubiquitous cell surface antigens Beta-2-Microglobulin and CD298 (Biolegend Totalseq A Hashtag antibodies). Hashed samples were pooled and stained together with a cocktail of 73 oligonucleotide-conjugated antibodies targeting cell surface antigens, including principal lineage antigens, that also includes isotype control antibodies (Biolegend Totalseq A). The stained pool of cells was washed, and non-viable cells removed using a magnetic Dead Cell Removal kit (StemCell). Each batch of pooled samples was then encapsulated across 5 lanes of a 10X Genomics v3.0 chip with a target cell capture of $3 \times 10^4$ cells per lane. An average of ~4600 cells (minimum 3844, maximum 6404) were encapsulated per IMRAS trial sample analyzed in this study.

Separate sequencing libraries were generated for gene expression (GEX), oligonucleotide-conjugated antibodies (ADT), and hashtags (HTO). Hashtag libraries were sequenced using the Illumina NextSeq 500/550 High Output (75 cycle) and NovaSeq S1 (100 cycle) kits at a target depth of 1,000 reads per cell, and gene expression and antibody libraries were sequenced using the Illumina NovaSeq S2 (100 cycle) kit at a target depth of 25,000 and 10,000 reads per cell, respectively. The binary base call (BCL) sequence files were base-called and demultiplexed using cellranger mkfastq (v3.1.0). Alignment to the human reference GRCh38 and feature counts were performed using cellranger count (v3.1.0). GEM wells for each sample were aggregated and depth normalized using cellranger aggr (v3.1.0).

## Unsupervised clustering of flow cytometry data

Pre-filtering and pre-gating using FlowJo was performed to select live single-cells from each sample, and pre-gating was performed on canonical markers for each panel (T-cell: CD3+-CD19-CD14-; B cell: CD19+CD3-; invariant T cell: CD3+CD14-CD19-; DC: CD3-CD19-; NK cell: CD14-CD19-CD3-). The invariant T cell and NK cell panel originate from the same sample data and antibody panel. Marker intensities for each sample were scaled using the biexponential scale prior to clustering with the Seurat FindNeighbours and FindClusters function. Unsupervised clustering was then carried out using the R Seurat [34] and flowCore (v2.6.0) [82] packages to identify distinct cell subsets separately for all 5 flow cytometry panels. Cluster count tables were output by scaling total per-cluster counts to total per-sample live pre-gated cells. Expression of markers in each cluster was used to manually annotate cell types (**S3 Fig and S3 Data**). Certain clusters were designated as "unidentified" due to incomplete full spectrum of protein markers in the antibody panels.

## Clustering of the CITE-seq data using weighted nearest neighbor (WNN) analysis

A Seurat (v4.0.1) [34,83] object was created for each sample following debris removal and barcode mapping to samples. Raw count data was pre-processed to remove low quality cells with high mitochondrial RNA fraction (25%), very high UMI counts (nCount_RNA >25,000), and very low or very high gene counts (nFeature_RNA <200 and >4000). The resulting per sample

RNA and ADT data were independently normalized using standard or center log normalization respectively, then each integrated across all samples using the canonical correlation analysis (CCA) method to correct for any batch effects and control for between-sample technical variation. To allow for the integration of multimodal datasets, RNA and ADT weighted similarities were used to construct the weighted nearest neighbor (WNN) graph and to perform WNN-based uniform manifold approximation and projection (UMAP) plot as described [84]. Manual annotation of cellular identity was performed by finding differentially expressed genes and ADTs for each cluster using Seurat's implementation of the Wilcoxon rank-sum test ("FindMarkers" function) and comparing those markers with known cell type–specific genes and surface marker proteins (S4 Fig and S4 Data).

## Statistical and bioinformatics analysis

To calculate differentially expressed genes (DEGs) over time points in the longitudinal data, linear mixed-effect regression model (LMER) was fit through the R package glmmSeq (v.0.1.0) [85]. Gene expression (CPM) was fit as a dependent variable while sample time point was fit as a fixed effect, and individual identifier as a random effect. Dispersion was calculated using the edgeR (v3.36.0) package function, estimateDisp. For DEG calculation on comparisons between sample groups; protected, non-protected, and mock, at each time point, the R package DESeq2 (v1.34.0) was used [86].

Gene set enrichment analysis (GSEA) was performed using the R package fgsea (v1.20.0) [87]. Genes were ranked by the average fold change across different time interval comparisons and separately for each sample groups. Normalized enrichment score (NES) for non-statistically significant gene sets were adjusted to 0. The blood transcriptomic module (BTM) gene sets [30,31] were obtained from tmod R package (v0.46.2) [88] while the MsigDB Hallmark gene set [29] was obtained from download site http://www.gsea-msigdb.org/gsea/msigdb/genesets.jsp?collection=H.

Cell deconvolution analysis from bulk RNA transcriptomic data was performed using the immunoStates matrix [33] as part of MetaIntegrator R package (v2.1.3) [89]. For Ingenuity Pathway Analysis (IPA) [32], content version 68752261 was used to generate Canonical Pathway enrichment analysis results.

For flow cytometry data, LMER was used to calculate cell subsets that significantly change over time points and were significantly different between group comparisons. The R lme4 package (v1.1–27.1) [90] was used to fit the model using the lmer function. Cell cluster frequency generated by the unsupervised clustering was fit as a dependent variable while sample time points and group were fit as fixed effects, and individual identifier as a random effect. The mixed-effect models were fit as follow:

Full model : Cell frequency ~ 1 + Time point * Group + (1|Subject ID)
Reduced model 1 : Cell frequency ~ 1 + Time point + (1|Subject ID)
Reduced model 2 : Cell frequency ~ 1 + Group + (1|Subject ID)

Comparisons between the full model and reduced models were evaluated to determine significant relationships between cell frequency and time point and group variables. Specifically, ANOVA was used to compare the full model and reduced model 2, in which the P values represent statistical significance of improved fit associated with time variable, i.e. response-associated cell subsets. Within the statistically significant response-associated cell subsets, full model and reduced model 1 were compared using ANOVA to calculate significantly different cell subsets between groups.

For single-cell RNA sequencing data, significant changes in cell subsets overtime and GSEA were analyzed as above. Only "reduced model 1" was used to fit the mixed model, followed by ANOVA to test for statistical significance of the fixed effect, i.e. time point variable.

The FDR-corrected P values were calculated using the Benjamini-Hochberg method. For DEG, deconvoluted cell proportion, and flow cytometry cell subset frequency data, the FDR threshold was set at <0.2. For GSEA (both bulk and single-cell transcriptomics), IPA, and scRNA-seq cell subset frequency data, the FDR threshold was set at <0.01.

## Supporting information

**S1 Table. Sample availability from the IMRAS trial.**
(DOCX)

**S2 Table. List of antibodies used for flow cytometry experiment.** Four different staining panels were used for the detection of T cell, NK and unconventional T cell, DC, and B cell.
(XLSX)

**S1 Data. List of cluster annotation on single-cell RNA sequencing data.** Top 20 oligonucleotide-conjugated antibodies (ADTs) and genes on each cluster.
(XLSX)

**S2 Data. Gene set enrichment analysis (GSEA) output from the whole blood RNAseq data.**
(XLSX)

**S3 Data. List of pathways generated by the Ingenuity Pathway Analysis (IPA) on comparison between C0 vs B0, C1 vs C0, and C7 vs C0.**
(XLSX)

**S4 Data. Cluster frequency on flow cytometry data for each antibody panel.**
(XLSX)

**S5 Data. List of cluster annotation on flow cytometry data for each antibody panel.**
(XLSX)

**S1 Fig. Cell subset clusters generated using weighted nearest neighbor approach on the CITE-seq data.** Cell clusters were generated using the Seurat package for weighted nearest neighbor (WNN) clustering on both RNA and ADT data. Each number indicates a distinct cluster with unique **A)** gene and **B)** protein expression profile.
(TIF)

**S2 Fig. Gene expression for TUBB2A, B4GALNT3, and TREML4 gene in each sample group over time.** Lines indicate each individual. Dashed lines indicate LOESS regression with 95% confidence interval shown in the highlighted color. Gene expression is in counts per million (CPM) P (n = 6) indicates PfRAS-vaccinated protected individuals, NP (n = 5) indicates PfRAS-vaccinated non-protected individuals, Mock (n = 3) indicates non-infected mosquito bites vaccinated individuals.
(TIF)

**S3 Fig. Gene set enrichment analysis (GSEA) in each sample group over time, ordered on enrichment score.** GSEA was analyzed on different comparisons between time points; C0 vs B0, C1 vs C0, C7 vs C0, for each sample group. FDR q-value cutoff was set at 0.01. Each row indicates a gene set module. Bar plots indicate normalized enrichment score (NES) and are colored according to high-level annotations of the gene set modules.
(TIF)

**S4 Fig. Gene set enrichment analysis (GSEA) transcriptomic response using the Hallmark gene sets.** GSEA was performed on comparisons between time points; C0 to B0, C1 to C0, C7 to C0, for each sample group. FDR q-value cutoff was set at 0.01. Each square indicates a gene

set module. Color represents normalized enrichment score (NES) obtained from GSEA. Colored row annotations represent high-level annotations of the gene set modules. P (n = 6) indicates PfRAS-vaccinated protected individuals, NP (n = 5) indicates PfRAS-vaccinated non-protected individuals, Mock (n = 3) indicates non-infected mosquito bites vaccinated individuals.
(PDF)

**S5 Fig. Leading-edge genes within significantly enriched interferon and NK cell-associated blood transcriptomic modules (BTMs).** A) Shared leading-edge genes at different time point comparisons for interferon and NK cell-associated BTMs. B) Gene expression levels. Lines indicate each individual. Dashed lines indicate LOESS regression with 95% confidence interval shown in the highlighted color. Gene expression is in counts per million (CPM). P (n = 6) indicates PfRAS-vaccinated protected individuals, NP (n = 5) indicates PfRAS-vaccinated non-protected individuals, Mock (n = 3) indicates non-infected mosquito bites vaccinated individuals.
(TIF)

**S6 Fig. Cell subset clusters generated from unsupervised clustering on flow cytometry data.** Cell clusters were generated using the Seurat package for unsupervised clustering on each antibody panels. Each number indicates a distinct cluster with unique marker expression profile. Flow cytometry standard (fcs) files were pre-gated to exclude non-relevant cells such as doublets and non-lymphocytes prior to clustering analysis. Additionally, panels were also pre-gated as follows; 1) T cell panel: CD14-CD19-CD3+, 2) B cell panel: CD3-CD19+, 3) invariant T cell panel: CD14-CD19-CD3+, 4) DC panel: CD19-CD3-, 5) NK panel: CD14-CD19-CD3-. The invariant T cell panel and the NK panel originate from the same fcs files and antibody panel.
(TIF)

## Acknowledgments

We would like to acknowledge all the IMRAS study participants who made this work possible. We thank the Naval Medical Research Center (NMRC) Malaria Department, the Clinical Trials Center, and the Clinical Immunology Laboratory staff led by Dr. Eileen Villasante, Dr. Judith Epstein, Dr. Martha Sedegah, respectively, that conducted the IMRAS clinical trial and processed, cryopreserved, inventoried and transferred the clinical samples. We would like to acknowledge the Research Scientific Computing at Seattle Children's Research Institute for providing HPC resources that have contributed to the research results reported within this paper. _FTC

## Author Contributions

**Conceptualization:** Damian A. Oyong, Kenneth D. Stuart.

**Formal analysis:** Damian A. Oyong, Fergal J. Duffy, Maxwell L. Neal, Ying Du, Seong-Hwan Jun, Helen Miller, Suzanne M. McDermott.

**Funding acquisition:** John D. Aitchison, M Juliana McElrath, Kenneth D. Stuart.

**Investigation:** Damian A. Oyong, Jason Carnes, Katharine V. Schwedhelm, Nina Hertoghs.

**Methodology:** Damian A. Oyong, Fergal J. Duffy, Maxwell L. Neal, Ying Du, Nina Hertoghs, Stephen C. De Rosa, Suzanne M. McDermott.

**Resources:** John D. Aitchison, Stephen C. De Rosa, Evan W. Newell, M Juliana McElrath.

**Supervision:** Kenneth D. Stuart.

**Validation:** Jason Carnes, Katharine V. Schwedhelm, Nina Hertoghs.

**Visualization:** Damian A. Oyong, Helen Miller.

**Writing – original draft:** Damian A. Oyong.

**Writing – review & editing:** Damian A. Oyong, Fergal J. Duffy, Maxwell L. Neal, Ying Du, John D. Aitchison, Evan W. Newell, M Juliana McElrath, Suzanne M. McDermott, Kenneth D. Stuart.

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
