## [Decision Letter · Decision Letter 0]

24 Jan 2023

Dear Dr. Stuart,

Thank you very much for submitting your manuscript "Distinct immune responses associated with vaccination status and protection outcomes after malaria challenge" for consideration at PLOS Pathogens. As with all papers reviewed by the journal, your manuscript was reviewed by members of the editorial board and by several independent reviewers. In light of the reviews (below this email), we would like to invite the resubmission of a significantly-revised version that takes into account the reviewers' comments.

All reviewers agreed that the study was well performed and important. But there were differences in opinion regarding the study's novelty and potential impact. I agree with this and suggest that some of the assays proposed by Reviewer #3 be performed.

We cannot make any decision about publication until we have seen the revised manuscript and your response to the reviewers' comments. Your revised manuscript is also likely to be sent to reviewers for further evaluation.

Sincerely,

Ira J Blader

Guest Editor

PLOS Pathogens

Margaret Phillips

Section Editor

PLOS Pathogens

Kasturi Haldar

Editor-in-Chief

PLOS Pathogens

orcid.org/0000-0001-5065-158X

Michael Malim

Editor-in-Chief

PLOS Pathogens

orcid.org/0000-0002-7699-2064

All reviewers agreed that the study was well performed and important. But there were differences in opinion regarding the study's novelty and potential impact. I agree with this and suggest that some of the assays proposed by Reviewer #3 be performed.

Reviewer's Responses to Questions

**Part I - Summary**

Reviewer #1: This manuscript undertakes a transcriptomic analysis of PBMCs from individuals that received irradiated sporozoite vaccination and Plasmodium falciparum challenge. The authors are commended for summarizing the data in easy to digest figures and overall I think this paper provides a useful resource for the malaria vaccine community. I may have missed it but if the authors are making the raw data available for access by other members of the community after publication please include the link to this.

It is a little hard through the results as presented but overall this manuscript has been more or less well put together in a logical fashion and I do not have any major issues with what has been presented. There are some points which need a little clarification (below).

Reviewer #2: This is a well-executed study of peripheral blood cell phenotype and gene expression changes during vaccination and challenge infection. The study design allows comparison of vaccinated/protected subjects with vaccinated/unprotected and mock vaccinated subjects; this is (I think) novel and allows the effects of vaccination to be separated from the effects of parasitemia during challenge. As far as I can tell (I am not an expert in some of the techniques employed) the data have been collected and analysed to a high standard. Data interpretation is measured and conclusions have not been overstated. Overall, the findings are in line with expectations from previous studies of system-wide responses to malaria vaccination and CHMI and where there are differences, they have been explored openly and with insight.

However, I do fear that we have reached the stage of diminishing returns with this type of study. The inherent limitations of the approach - most significantly that studying peripheral blood tells us very little about the responses we are really interested in (which, in malaria infections, are likely happening mostly in the spleen and liver) and this really limits our ability to move from observation to understanding. The authors are aware of this, and address the issue directly in their discussion, but it really does limit the conclusions that we can draw.

For example, the authors state that “Prior to CHMI but after vaccination, immune profiles were relatively similar between the two PfRAS vaccinee groups, protected and non-protected” . So, their data cannot distinguish protective vs non protective responses to vaccination? This suggests (as alluded to above) that the protective responses induced by vaccination are either not cellular/gene expression related or are not manifest in blood. Similarly, post CHMI responses "differ between protected and non-protected/mock" – i.e. they reflect the course of infection rather than the impacts of vaccination? So - which markers are indicative of a protective vaccine-induced response?

We now need to take the very many hypotheses that are emerging from such studies and test them experimentally.

Reviewer #3: The manuscript by Oyong et al., sought to study the immune mechanisms of protection following vaccination with radiation-attenuated Plasmodium falciparum sporozoites (PfRAS) in a controlled human malaria infection (CHMI) setting. Transcriptional profiling of whole blood and PBMC profiling via high-dimensional flow cytometry analysis was performed on volunteers who received either PfRAS or noninfectious mosquito bites, followed by CHMI. Approximately half of the PfRAS-vaccinated volunteers were protected from blood stage infection following CHMI, allowing for stratification of data based on protection status. The authors identified various gene sets, blood transcriptional modules, and immune pathways that were differentially expression or regulated at different timepoints between the groups (mock vaccinated, protected, and not protected). For example, gene sets associated with interferon responses were increased in protected vaccinees one day following CHMI. The authors performed a considerable amount of data analysis for this manuscript. However, overall, the data are observational, correlative, and not novel (largely confirm data in the literature). More specifically, no functional or mechanistic studies were performed to confirm the observational data. Additionally, while the authors sought to determine correlates of protection, the analyses were so high-level that it was difficult to determine what could possibly be related to protection. (The lack of functional and mechanistic studies also hindered that goal.) Lastly, the analyses largely confirmed existing data in the literature, which strengthens the already published findings, but makes this study less novel. Focusing on certain aspects of the data such as the innate (NK, pDC, monocyte) data or the adaptive (T cell and B cell) data and performing a deeper dive into the existing data along with functional studies would make the study more novel, meaningful, and impactful.

**Part II – Major Issues: Key Experiments Required for Acceptance**

Reviewer #1: None

Reviewer #2: None

Reviewer #3: • While the manuscript contains a very large set of data, the data are observational and correlative. No mechanistic studies were performed to determine if the observed increase or decrease of certain pathways or gene signatures had functional relevance. For example, to confirm the relevance of increased interferon signaling pathways, IFN-a/b and IFN-g could be quantified in the plasma of study participants. Additionally, functional assays on immune cells could be performed, such as stimulating PBMCs with infected RBCs or PMA/I and quantifying cytokines in the supernatant as well as staining T cells and NK cells for cytokines (eg., IFN-g, TNFa, etc).

• Similarly, major findings from the RNA-seq data set should be confirmed by analysis of proteins using other methods (flow cytometry, ELISAs, etc). See bullet point above for analysis of plasma for cytokines and PBMC stimulation assays.

• The study sought to identify correlates of protection, but there appeared to be little emphasis on identifying differences at C0 between the mock, protected and non-protected groups. Data in Fig 3B showed that increases in interferon and NK cell-associated responses were observed in protected vaccinees. A deeper dive into the genes/pathways in those BTMs could provide information on targets for correlates of protection studies or pathways/genes to target for vaccination strategies to boost protection.

• The amount of data included in this manuscript is incredible and the analysis is impressive. However, the descriptions of the data and overall conclusions were high-level. While I can see the importance of keeping all of these data together in one manuscript, it makes it difficult to draw any real meaning. It might be more useful to focus on certain aspects of the data to make functional – and ideally mechanistic – conclusions. For instance, the NK cell data are interesting, along with the pDC, monocyte and interferon data – perhaps a focus on the innate immune response? Or, the T cell and B cell data are interesting, too, especially in relation to the connection with gamma delta T cells and “atypical memory B cells” seen in other malaria studies. Perhaps a deeper dive into the existing data along with some additional functional studies could provide a clearer picture of what’s going on between the mock-vaccinated, protected and not protected groups.

**Part III – Minor Issues: Editorial and Data Presentation Modifications**

Reviewer #1: I think figure 1 would benefit from a visual timeline of the B and C time points – it is explained in the methods and included in Fig S1 but such a figure would be a useful addition to help with interpretation of the figure and I would move figure S1 to the main figure.

In the figure in general some of the panels are hard to read as the writing is incredibly small. It would help to try and increase font sizes where possible. (applies particularly to Fig. 1B, Fig. 4A, Fig S4, Fig. S6.

In Figure 1 the mock control says 4 of the 5 individuals were selected Why are 4 selected? On what basis?

In Figure 4 I would change the titles to all be in order (B0 vs C0, C0 vs C1 and C0 vs C7). It is a small point but helps to order the time points for interpretative purposes.

Discussion

Lines 419-420 It may not be a decreased expression of adaptive immunity. What did the adaptive immune response look like? Although the population sizes are different proportionally, is the gene expression also different? Proportion can be driven by expansion of non T/B cells.

Line 431: Malaria infection does not exist - malaria is a disease caused by Plasmodium infection. Also malaria parasites (Line 495) – it is Plasmodium parasites.

Reviewer #2: Cell frequencies can be misleading - an increase in one subset could be a genuine expansion of that subset or a selective loss of other subsets. Is it possible to calculate (or approximate) absolute cell numbers using CBC data?

The authors claim - in the discussion that their study is “providing important insights into immune mechanisms required to achieve sterilizing immunity” – what exactly are these important insights?

Reviewer #3: • Reference to “interferon responses” is made numerous times, but it is often unclear if it is referring to type I IFN, type II, etc., or if that cannot be determined from the dataset. Either way, that should be clarified.

• The text in the IPA dataset shown in Fig 4A is nearly impossible to read (had to zoom in to 300% on PDF). The text size should be increased or the data should be shown in a more easily-readable format like a heatmap.

• While this type of study is important, the relevance to vaccination efforts in malaria-endemic countries is unclear since it’s known that Plasmodium vaccinations have much lower efficacy rates in malaria-endemic countries. Could the authors could make this connection by comparing their RNA-seq data to published RNA-seq data from studies of malaria patients or vaccination studies in endemic countries?

• It is unclear how much the immune profile (transcriptional profile, cellular make-up, etc) in the blood can be used to understand tissue-specific (ie, liver) immune responses. In other words, more data comparing the liver-specific immune response to the circulating immune response – in non-human primate or small animal models – are needed before these types of datasets have much meaning.

PLOS authors have the option to publish the peer review history of their article (what does this mean?). If published, this will include your full peer review and any attached files.

Reviewer #1: No

Reviewer #2: No

Reviewer #3: **Yes: **Kristina S. Burrack
---

## [Decision Letter · Decision Letter 1]

27 Mar 2023

Dear Dr. Stuart,

Thank you very much for submitting your manuscript "Distinct immune responses associated with vaccination status and protection outcomes after malaria challenge" for consideration at PLOS Pathogens. As with all papers reviewed by the journal, your manuscript was reviewed by members of the editorial board and by several independent reviewers. The reviewers appreciated the attention to an important topic. Based on the reviews, we are likely to accept this manuscript for publication, providing that you modify the manuscript according to the review recommendations.

Reviewer #2's editorial suggestions are valid and should be incorporated

Sincerely,

Ira J Blader

Guest Editor

PLOS Pathogens

Margaret Phillips

Section Editor

PLOS Pathogens

Kasturi Haldar

Editor-in-Chief

PLOS Pathogens

orcid.org/0000-0001-5065-158X

Michael Malim

Editor-in-Chief

PLOS Pathogens

orcid.org/0000-0002-7699-2064

Reviewer #2's editorial suggestions are valid and should be incorporated

Reviewer Comments (if any, and for reference):

Reviewer's Responses to Questions

**Part I - Summary**

Reviewer #1: The authors have addressed my comments. I think the point was raised that this study is not mechanistic and the limitation of studying PBMCs was raised. Whilst I agree it is not mechanistic and PBMCs give limited information, my personal opinion is that this data represents a valuable and searchable resource for the malaria community. We need to be able to do more mechanistic studies, but we also need to be able to extrapolate form human studies (ie. PBMCs). Yes it confirms some of the existing literature but I think this validates the dataset which can be further mined by the community as new angles of research occur.

Reviewer #2: The authors have edited the manuscript to improve the clarity of both the text and the figures, and I thank them for that.

The major limitations of the study remain, however, and there are no samples remaining that might allow any experimental exploration of their observations. The decision whether or not to accept the ms for publication is thus down to editorial judgement: are the data presented sufficiently novel and sufficiently useful to others to warrant publication in PLoS Pathogens? I am on the fence about this.

One observation however: the conclusion I draw from the authors responses to some of the reviewer comments is that the very early (within 24h) innate immune response influences the outcome of CHMI and this early response is heterogeneous between subjects both after immunisation but also, crucially, before immunisation (or in the absence of immunisation). In other words, underlying heterogeneity in innate immune responses (possibly linked to NK/KIR, but this is not proven) affects the outcome of CHMI and may also affect the outcome of immunisation. I am reminded of a paper published almost 20 years ago (Walther et al, Immunity 2005 Sep;23(3):287-96. doi: 10.1016/j.immuni.2005.08.006.) that came to similar conclusions - that the immediate innate response drives either control of parasitemia or control of the fever response after CHMI - and I wonder whether this conclusion needs to be stated even more clearly in the abstract and discussion of this present manuscript? The relevance of this observation is not only that it may explain the heterogeneous response to vaccination in the US volunteers but may also underlie population level differences in vaccine efficacy?

Reviewer #3: The revised manuscript by Oyong et al., describes immune mechanisms of protection following vaccination with radiation-attenuated Plasmodium falciparum sporozoites (PfRAS) in a controlled human malaria infection (CHMI) setting. Transcriptional profiling of whole blood and PBMC profiling via high-dimensional flow cytometry analysis was performed on volunteers who received either PfRAS or noninfectious mosquito bites, followed by CHMI. Approximately half of the PfRAS-vaccinated volunteers were protected from blood stage infection following CHMI, allowing for stratification of data based on protection status. The authors addressed this reviewer’s concerns adequately and provided additional analyses that strengthen their conclusions. Overall, this manuscript includes important data that will shed light on immune mechanisms of protection following whole sporozoite Plasmodium vaccination.

**Part II – Major Issues: Key Experiments Required for Acceptance**

Reviewer #1: (No Response)

Reviewer #2: Amend abstract and discussion to really stress the pivotal role of early innate responses (and the heterogeneity therein) in the outcome of CHMI and vaccination.

Reviewer #3: none

**Part III – Minor Issues: Editorial and Data Presentation Modifications**

Reviewer #1: (No Response)

Reviewer #2: (No Response)

Reviewer #3: none

PLOS authors have the option to publish the peer review history of their article (what does this mean?). If published, this will include your full peer review and any attached files.

Reviewer #1: No

Reviewer #2: No

Reviewer #3: **Yes: **Kristina S. Burrack

Figure Files:

Data Requirements:

Reproducibility:

References:

---

## [Editor Report · Decision Letter 2]

26 Apr 2023

Dear Dr. Stuart,

We are pleased to inform you that your manuscript 'Distinct immune responses associated with vaccination status and protection outcomes after malaria challenge' has been provisionally accepted for publication in PLOS Pathogens.

Best regards,

Ira J Blader

Guest Editor

PLOS Pathogens

Margaret Phillips

Section Editor

PLOS Pathogens

Kasturi Haldar

Editor-in-Chief

PLOS Pathogens

orcid.org/0000-0001-5065-158X

Michael Malim

Editor-in-Chief

PLOS Pathogens

orcid.org/0000-0002-7699-2064
---

## [Editor Report · Acceptance letter]

9 May 2023

Dear Dr. Stuart,

We are delighted to inform you that your manuscript, "Distinct immune responses associated with vaccination status and protection outcomes after malaria challenge," has been formally accepted for publication in PLOS Pathogens.

Best regards,

Kasturi Haldar

Editor-in-Chief

PLOS Pathogens

orcid.org/0000-0001-5065-158X

Michael Malim

Editor-in-Chief

PLOS Pathogens

orcid.org/0000-0002-7699-2064